# Extract of *Artemisia dracunculus* L. Modulates Osteoblast Proliferation and Mineralization

**DOI:** 10.3390/ijms241713423

**Published:** 2023-08-30

**Authors:** Matthew C. Scott, Aleah Bourgeois, Yongmei Yu, David H. Burk, Brenda J. Smith, Z. Elizabeth Floyd

**Affiliations:** 1Pennington Biomedical Research Center, Baton Rouge, LA 70808, USA; matthew.scott@pbrc.edu (M.C.S.); yongmei.yu@uth.tmc.edu (Y.Y.); david.burk@pbrc.edu (D.H.B.); 2Department of Obstetrics and Gynecology, Indiana University School of Medicine, Indianapolis, IN 46202, USA; bsm14@iu.edu

**Keywords:** bone, osteoblast, MC3T3-E1, osteogenesis, botanicals, *Artemisia dracunculus* L., plant-based, PMI5011, DMC-2, Russian Tarragon, mineralization

## Abstract

Thiazolidinediones (TZD) significantly improve insulin sensitivity via action on adipocytes. Unfortunately, TZDs also degrade bone by inhibiting osteoblasts. An extract of *Artemisia dracunculus* L., termed PMI5011, improves blood glucose and insulin sensitivity via skeletal muscle, rather than fat, and may therefore spare bone. Here, we examine the effects of PMI5011 and an identified active compound within PMI5011 (2′,4′-dihydroxy-4-methoxydihydrochalcone, DMC-2) on pre-osteoblasts. We hypothesized that PMI5011 and DMC-2 will not inhibit osteogenesis. To test our hypothesis, MC3T3-E1 cells were induced in osteogenic media with and without PMI5011 or DMC-2. Cell lysates were probed for osteogenic gene expression and protein content and were stained for osteogenic endpoints. Neither compound had an effect on early stain outcomes for alkaline phosphatase or collagen. Contrary to our hypothesis, PMI5011 at 30 µg/mL significantly increases osteogenic gene expression as early as day 1. Further, osteogenic proteins and cell culture mineralization trend higher for PMI5011-treated wells. Treatment with DMC-2 at 1 µg/mL similarly increased osteogenic gene expression and significantly increased mineralization, although protein content did not trend higher. Our data suggest that PMI5011 and DMC-2 have the potential to promote bone health via improved osteoblast maturation and activity.

## 1. Introduction

Obesity increases the risk of developing multiple comorbidities including cardiovascular diseases, some cancers, peripheral insulin resistance, and type 2 diabetes (T2DM) [1]. Adding to this list is a growing appreciation of the impact of overweight and obesity on bone health. Until recently, obesity was considered beneficial to maintaining healthy bones. Individuals with a BMI categorized as overweight or obese have higher bone mineral density (BMD) when measured via dual energy X-ray absorptiometry (DEXA) [2]. Nevertheless, when BMI is combined with measures of visceral adiposity, the relationship between obesity and bone health becomes more complex. Higher levels of abdominal visceral fat accumulation are associated with reduced femoral neck BMD [3] and increased risk of fracture [4], including upper leg fractures in postmenopausal women [5]. Independent of BMD, fracture risk with obesity is elevated due to metabolic consequences related to peripheral insulin resistance and low-grade inflammation in the visceral adipose tissue [6]. 

Treatments for obesity-related insulin resistance and T2DM that focus on adipose tissue illustrate the complicated relationship between obesity and bone health. The anti-diabetic thiazolidinediones (TZDs) improve insulin sensitivity by activating peroxisome proliferator-activated receptor gamma (PPARγ), a pivotal nuclear receptor in adipogenesis and regulating glucose and lipid metabolism [7]. However, PPARγ’s action in adipogenesis also inhibits Wnt-signaling and the major osteogenic transcription factor Runx2.; TZD-dependent PPARγ activation causes bone loss by suppressing osteoblastogenesis [8]. Consequently, TZDs increase fat mass and the risk of osteoporosis by targeting PPARγ and adipose tissue to improve insulin sensitivity [9,10]. 

Nonetheless, a benefit of TZD-mediated treatment of T2DM is enhanced insulin responsiveness in skeletal muscle [11,12], the major site of insulin-dependent glucose uptake in peripheral tissues [13]. There are current efforts to develop partial PPARγ agonists that modulate skeletal muscle glucose uptake without weight gain, but the impact on bone health when treating T2DM or insulin resistance remains a concern [14,15].

*Artemisia dracunculus*, or Russian Tarragon, is a botanical with a history of culinary and medicinal uses. In studies using a rodent model of diet-induced obesity and insulin resistance, we found that an ethanolic extract from *A. dracunculus* (termed PMI5011) lowers blood glucose and improves insulin levels comparable to troglitazone [16] (a TZD) by enhancing insulin action in skeletal muscle [17]. In primary human skeletal muscle cell cultures from individuals with T2DM, PMI5011 increases glucose uptake, stimulates insulin-mediated signaling, and reduces inflammation [18,19,20,21]. PMI5011 also reduces lipid accumulation in skeletal muscle and hepatic tissue while improving responsiveness to carbohydrates or fats as a fuel source in skeletal muscle [22], potentially via activation of AMP-activated Protein Kinase (AMPK) [23]. Recent studies confirm that 2′,4′-dihydroxy-4-methoxydihydrochalcone (DMC-2, 4-O-methyldavidigenin) is the bioactive compound that improves insulin signaling in the skeletal muscle of obese, insulin-resistant mice [21]. PMI5011 or DMC-2 lowers blood glucose via a mechanism that alters insulin signaling in skeletal muscle, but not liver or adipose tissue, suggesting skeletal muscle is the primary target tissue. This raises the possibility that PMI5011-mediated improvements in insulin sensitivity and glucose homeostasis occur without compromising bone health. In the current study, we analyzed the effect of PMI5011 on adipose tissue expansion combined with in vitro studies of PMI5011 and DMC-2’s effect on osteogenesis using the MC3T3-E1 pre-osteoblast cell line to test our hypothesis that PMI5011 is bone sparing while not modulating adipose tissue.

## 2. Results

To determine if PMI5011 alters adipose tissue expansion, we analyzed multiple parameters of adipose tissue function in mice fed an obesogenic high-fat diet alone or supplemented with PMI5011 (1.2% *w*/*w*). We focused on adipose tissue in female mice because of the detrimental effect on bone health in women treated with insulin sensitizers that promote adipose tissue expansion via stimulating PPARγ activity [9,24]. Supplementation with PMI5011 did not alter body weight (Figure 1A) or fat mass (Figure 1B). Similarly, we observed no differences in the relative abundance of adiponectin (*Adipoq)*, leptin (*Lep)*, or peroxisome proliferator-activated receptor gamma (*Pparg*) gene expression (Figure 1C) or PPARγ protein (Figure 1D,E) as indicators of adipogenesis. Adipose tissue morphology indicated PMI5011 is associated with a trend toward larger fat cells with pockets of multilocular fat cells consistent with beneficial beiging of the subcutaneous fat (Figure 1F). Even so, fat cell size was unchanged (Figure 1H), and the isolated regions of beiging were not associated with alterations in the beiging markers of indoleamine 2,3-diooxygenase (*Dio*), uncoupling protein-1 (*Ucp1*), and PPARγ coactivator-1 alpha (*Pgc1a*) in the whole tissue (Figure 1I).

As bone was not collected from the animal cohort, we used a murine pre-osteoblast cell line, MC3T3-E1 subclone 4, to test osteogenesis by examining differentiation with and without botanical treatment. First, we confirmed the previously reported efficacy of the MC3T3-E1 model system (Figure 2) [25]. Cells treated with 5 mM beta-glycerol phosphate and 0.2 mM ascorbic acid (differentiated) were compared to cells grown in media without ascorbic acid (MEMα–AA, non-differentiated). The relative abundance of genes, encoding proteins with a range of functions that support osteogenesis, was increased from day 1 to 12 in cells induced toward osteoblast differentiation compared to non-induced cells: runt-related transcription factor 2 (*Runx2*), osterix (*Sp7*), osteocalcin (*Bglap*), integrin-binding sialoprotein (*Ibsp*), osteopontin (*Spp1*), and dentin matrix protein 1 (*Dmp1*) (Figure 2A). Further, differentiated cells showed higher expression of Runx2 and β-catenin proteins, early regulators of osteoblastogenesis [26] at day 6 post induction (Figure 2B). Lastly, staining for alkaline phosphatase (Alk Phos, day 6) [27], collagen (day 6, Picro Sirius) [28], and mineralization (Von Kossa, day 12) [29] were increased in the differentiated compared to the non-differentiated cells (Figure 2C). 

To evaluate PMI5011’s effect on MC3T3-E1 cells, we first assayed cell viability using an MTT assay by comparing undifferentiated cells treated with increasing concentrations of PMI5011 (3 µg/mL, 10 µg/mL, and 30 µg/mL) to vehicle-treated cells (DMSO) (Figure 3A). MTT assay absorbance was unaffected by PMI5011 at 3 and 10 µg/mL. However, cells treated with 30 µg/mL PMI5011 showed a significant decrease in absorbance at 570 nm (−12%) compared to cells treated with DMSO alone. The 12% decrease in absorbance we observed with 30 µg/mL PMI5011 indicates fewer viable cells but cannot differentiate between cytotoxicity, proliferation, or apoptosis [30,31]. However, both proliferation and apoptosis are part of the osteogenic differentiation process [32,33,34]. To assay the effects of PMI5011 on proliferation and apoptosis during differentiation, we measured *Ki67* mRNA levels as a marker of proliferation [35] and *Casp3* mRNA as a marker of apoptosis [36,37] in cells post differentiation at day 1, 3, 6, and 12 (Figure 3B). Day 1 post induction was associated with the highest levels of *Ki67* and *Casp3* mRNA expression, independent of PMI5011 supplementation at 3, 10, or 30 µg/mL but significantly reduced by PMI5011 at 30 µg/mL when compared to 0 µg/mL. As differentiation progressed (day 3), lower concentrations of PMI5011 (3 and 10 µg/mL) reduced *Ki67* and *Casp3* mRNA. Although PMI5011 supplementation had no effect on *Ki67* mRNA levels at later stages of osteogenesis, PMI5011 at 10 or 30 µg/mL stimulated *Casp3* mRNA expression compared to differentiation alone at day 6. Together with the MTT assay, changes in *Ki67* and *Casp3* mRNA expression during differentiation predict that PMI5011 modulates proliferation and apoptosis independently during osteogenesis.

Next, we examined the effect of PMI5011 on outcomes related to osteogenic differentiation. We began by examining genes encoding two central transcription factors Runx2 and Sp7 for osteoblastogenesis. PMI5011 at 30 µg/mL increased Runx2 (day 6 and day 12) and Sp7 mRNA (day 6) at later stages of differentiation (Figure 4A). Differences in gene expression with PMI5011 were also associated with increased Runx2 protein expression by day 6 post induction but not Sp7 (Figure 4B). The major osteogenic Wnt/β-catenin signaling pathway converges with other signaling pathways to influence Runx2-stimulated osteoblastogenesis [38,39]. The stability of β-catenin protein is central to that pathway, but β-catenin steady-state levels were not altered with PMI5011 (Figure 4B), suggesting PMI5011 affects Runx2 expression independently of the Wnt/β-catenin pathway. To determine if the PMI5011-mediated changes in Runx2 and Sp7 mRNA corresponded with increased expression of their target genes, we assayed the gene expression of transcripts that encode a family of non-collagenous extracellular matrix proteins (Bglap, Ibsp, Spp1, and Dmp-1) that are associated with bone mineralization. Gene expression for *Bglap*, *Ibsp*, *Spp1*, and *Dmp1* was significantly increased on day 6 for cells induced with 30 µg/mL PMI5011, and by day 12, significant increases were found with all doses of PMI5011. Lastly, we evaluated alkaline phosphatase (Alk Phos) and collagen (Picro Sirius) on day 6 post induction and calcium depositions (Von Kossa) on days 10 and 12 post induction. Alkaline phosphate staining was not significantly increased by induction of osteogenesis under control or PMI5011-treated conditions on day 6, and PMI5011 did not alter collagen accumulation in differentiated cells on day 6 (Figure 4D,E). However, on days 10 and 12, mineralization trended upward with 10 µg/mL and 30 µg/mL PMI5011 based on Von Kossa staining compared to the cells differentiated in the absence of PMI5011 (Figure 4D,E). 

Because our experiments indicate PMI5011 modulates osteoblast differentiation (Figure 4), we tested a bioactive compound in PMI5011, 2′,4′-dihydroxy-4-methoxydihydrochalcone (4-O-methyldavidigenin, DMC-2), that was shown previously to enhance insulin signaling in skeletal muscle via stimulation of protein kinase B (AKT) phosphorylation [21]. For our DMC-2 experiments, we used 1 µg/mL of synthetic DMC-2 [40], as DMC-2 comprises less than 3% of PMI5011 by weight [21], and our most robust outcomes for PMI5011’s effect on osteogenesis was observed at 30 µg/mL (Figure 4). We used a similar methodology for determining the effects of 1 µg/mL DMC-2 on osteoblast differentiation and activity (Figure 5) as was used for PMI5011. DMC-2 treatment led to a small but significant increase in *Ibsp* on day 6 post induction, and on day 12 post induction, there are consistent increases in mRNA levels for osteogenic genes with substantial increases in *Spp1* and *Dmp1* (Figure 5A,B), similar to observations with the total extract PMI5011 (Figure 4A,B). We next examined the effects of 1 µg/mL DMC-2 on osteogenic proteins where SP7 trends higher on day 1 post induction, but β-catenin, Runx2, and SP7 protein content were reduced in cell lysate collected on day 6 (Figure 5C). Like PMI5011, DMC-2 did not affect staining for alkaline phosphatase and collagen (Figure 5D,E). Von Kossa staining, however, indicated a more robust and significant increase in mineralization on day 14 and 16 in the presence of DMC-2 compared to those differentiated without treatment (Figure 5D,E). 

## 3. Discussion

There is a rich history of using botanicals and their extracts to improve bone health [41,42]. Even so, little is known about the potential of botanically based dietary supplements to improve obesity-related insulin resistance while protecting bone health. The current approaches to improving insulin sensitivity in overweight and obese adults target adipose tissue, either by expanding fat mass by stimulating new fat cell formation [7,10,43] or by reducing fat mass using caloric restriction [44]. Both approaches have detrimental effects on bone health [9,45] and underscore the need for alternative approaches to managing obesity-related metabolic disorders that preserve bone health. An ethanolic extract from *A. dracunculus*, termed PMI5011, lowers blood glucose and maintains muscle mass without lowering body weight in animal models of obesity-related insulin resistance and type 2 diabetes (T2DM) [16,46]. We hypothesized that PMI5011 improves glucose homeostasis without harming bone quality as it acts in skeletal muscle, not adipose tissue [21]. 

Our results here show that PMI5011 supplementation does not alter body weight or stimulate fat mass expansion in female mice fed an obesogenic, high-fat diet. These results combined with previous reports support PMI5011′s potential to improve insulin sensitivity in obesity [16] without harming bone health related to fat mass expansion [47]. To begin exploring the effect of PMI5011 on bone cells, we used MC3T3-E1 pre-osteoblasts. Rather than simply bone sparing, our study shows PMI5011 has positive effects on some of our experimental outcomes for osteogenesis. The PMI5011-treatment-mediated up-regulation of *Runx2* and *Sp7*, major pro-osteogenic transcription factors, is associated with increased expression of Sp7 transcription factor-targeted genes (*Dmp1*, *Spp*/*Opn*, and *Ibsp*) encoding a subset of small integrin-binding ligand N-linked glycoproteins (SIBLING) and *Bglap* (osteocalcin). Both the SIBLING proteins and osteocalcin are non-collagenous products of osteogenesis necessary for extracellular matrix mineralization during bone formation [48]. The potential for PMI5011-mediated bone mineralization by modulating a subset of SIBLING proteins and osteocalcin is further supported by our morphological data showing maintenance of alkaline phosphatase and collagen levels and trends toward increased Von Kossa staining.

Mineralization in the MC3T3-E1 osteogenic model is significantly increased by 2′,4′-dihydroxy-4-methoxydihydrochalcone (DMC-2), a bioactive found in PMI5011. DMC-2 has been reported to account for the insulin-sensitizing effects of PMI5011 by stimulating phosphorylation of Protein kinase B (AKT) in skeletal muscle [21] without the AMP-activated protein kinase (AMPK) modulating effects of PMI5011 [23]. The enhanced effect of DMC-2 on mineralization points to a potential mechanism of action of PMI5011 and DMC-2 in osteogenesis, as both Akt [49,50] and AMPK [51,52] have roles in osteoblast maturation and function. Although not shown in the current study, but consistent with DMC-2 enhanced AKT stimulation in skeletal muscle and the role of Akt in bone formation [53], we anticipate DMC-2 promotes bone formation by enhancing insulin action in osteoblasts, similar to its mechanism of action in skeletal muscle. 

Given the strong structural and functional relationship between bone and skeletal muscle as major components of the musculoskeletal system, a shared mechanism of action of PMI5011 and DMC-2 in skeletal muscle and bone to regulate insulin sensitivity would not be surprising. Approximately 80% of insulin-dependent glucose uptake in peripheral tissues occurs in the skeletal muscle [54]. Bone is increasingly appreciated for its role in systemic glucose metabolism, and there is substantial evidence of cross-talk between skeletal muscle and bone [55,56,57]. As an example, bone-derived osteocalcin contributes to bone mineralization [58] while also acting as an osteokine to promote glucose uptake in skeletal muscle [59], maintain skeletal muscle mass with aging [57], and mediate glucose homeostasis [60]. Thus, positive modulation of bone or skeletal muscle health suggests favorable outcomes for both tissues and systemic insulin sensitivity. 

Considering the interaction between skeletal muscle and bone, botanicals that positively modulate both tissues to improve insulin sensitivity could be a substantial boon in treating obesity-induced insulin resistance despite, or perhaps due to, having no effects on adipose tissue. In this study, we demonstrated the potential for both PMI5011 and the bioactive DMC-2 to positively modulate bone formation. Our results indicating the potential of PMI5011 and DMC2 as bone-sparing botanicals in treating obesity-induced insulin resistance are novel but limited in scope without confirmation using primary bone cell cultures and extending the study to examining the impact of PMI5011 and DMC-2 on bone formation, in vivo, as well as bone resorption, in vitro and in vivo. Nonetheless, the prevalence of obesity, T2DM, osteoporosis, and the interactions between fat, muscle, and bone tissue indicate a need for alternative approaches to treating obesity-induced insulin resistance. Both PMI5011 and DMC2 improve insulin sensitivity via action in skeletal muscle, the major insulin-dependent glucose-utilizing organ in humans, and should be examined further for their potential to preserve bone quality while enhancing insulin sensitivity. 

## 4. Materials and Methods

### 4.1. PMI5011 and DMC-2 Extracts

The PMI5011 ethanolic extract from *A. dracunculus* was provided by the Botanical and Dietary Supplement Research Center at Pennington Biomedical Research Center. The PMI5011 ethanolic extract used in the current study is from the same lot as previously described [21]. The bioactive DMC-2 is 1.76% (*w*/*w*) of the total extract.

Detailed information about quality control, preparation, and biochemical characterization was reported previously [20,23,24,25,26]. Synthetic DMC-2 (2′, 4′–dihydroxy-4-methoxydihydrochalcone) was produced via custom synthesis to 99% purity by Gateway Biochemical Technology, Inc. as previously noted [40]. 

### 4.2. Animal Experiment for Adipose Tissue

Reproductively intact four-week-old female C57BL/6J mice were obtained from Jackson Laboratories (Bar Harbor, ME, USA). All animal experiments were approved by the Pennington Biomedical Research Center Animal Care and Use Committee (protocol #1011). The animals were singly housed with a 12 h light–dark cycle at 24 °C. At four weeks of age, mice of similar body weight were randomly assigned (n = 12/group) to a defined low-fat diet (LFD; 10% kcal fat and 17% kcal sucrose; Research Diets, New Brunswick, NJ, USA, #D12450H) or a low-fat diet supplemented with 1% *w*/*w* PMI5011 (Research Diets, custom diet #D15120050). After 4 weeks, the mice were switched to an HFD (45% kcal fat, Research Diets, #D12451) or the HFD supplemented with PMI5011, formulated with the same mass PMI5011/kcal (Research Diets, custom diet # D15102006; equivalent to 1.2% *w*/*w* HFD) as contained in the LFD and were fed ad libitum for 3 months. At the end of the study, mice were weighed, and nuclear magnetic imaging (Bruker, Billerica, MA, USA) was used to determine body composition (i.e., % fat). After fasting for 4–5 h, the mice were euthanized between 7 and 11 a.m. Body weight, body composition, and metabolic outcomes related to skeletal muscle and liver were previously reported [61]. Gonadal (gWAT) and subcutaneous (iWAT) adipose tissue were harvested and processed for H&E staining and protein and gene expression.

### 4.3. MC3T3-E1 Cell Culture

A murine pre-osteoblast cell line, MC3T3-E1 sub-clone 4 (American Type Culture Collection, Manassas, VA, USA) was used to examine in vitro osteogenesis with and without botanical supplementation. The cells were maintained in α-MEM (Gibco, Billings, MT, USA) with no ascorbic acid (MEMα–AA) supplemented with 10% FBS and 1% penicillin–streptomycin (P/S). For experimental cultures, the cells were plated in a 24-well format at 2 × 10^4^ cells per well in MEMα–AA, 10% FBS, 1% P/S. Cells were induced to differentiate once they reached ~100% confluence using induction media: MEMα, 10% FBS, 1% P/S, 5 mM beta-glycerol phosphate (βGP), and 0.2 mM ascorbic acid (AA). For PMI5011 and DMC-2 treatment conditions, induction media were supplemented with 3, 10, or 30 µg/mL PMI5011, 1 µg/mL DMC-2, or DMSO as the vehicle at 0.33% *v*/*v*. The cells were maintained in the treatments 0, 1, 3, 6, or 12 days to assess transcriptional changes and 0, 1, or 6 days to assess proteins of interest. After 6 days of treatment, collagen and alkaline phosphatase staining was performed and on 10–16 of treatment, Von Kossa staining was performed.

### 4.4. Gene Expression

Total RNA was isolated from powdered subcutaneous white adipose tissue (iWAT) or MC3T3-E1 cell monolayers using TRI Reagent (Molecular Research Center Inc., Cincinnati, OH, USA) and RNeasy Mini kits (Qiagen, Germantown, TN, USA) according to the manufacturers’ instructions. Isolated RNA (500 ng) was reverse transcribed using MultiScribe Reverse Transcriptase (Applied Biosystems, Foster City, CA, USA) with random primers at 37 °C for 2 h. Real-time PCR was performed with SYBRGreen Master Mix (Applied Biosystems) on the QuantStudio 6 Real-Time PCR System (Applied Biosystems) set to a comparative Ct with meltcurve. The assays were performed in triplicate, and the results were normalized to Cyclophilin B (iWAT) or ribosomal protein L13a (cell culture) and analyzed using the 2^−∆∆CT^ method with control group *lep* or *dio2* (iWAT), or day 0 non-differentiated (cell culture) values were used as the calibrator. The gene list and primers are provided in Table 1.

### 4.5. Protein Expression

Powdered adipose tissue was homogenized in a non-denaturing buffer containing 10 mM Tris-Cl, pH 7.4, 150 mM NaCL, 1 mM EDTA, 1 mM EGTA, 1% Triton X-100, 0.5% Igepal with protease inhibitors (1 µM PMSF, 1 µM pepstatin, 1 µg/mL leupeptin, 5 µg/mL aproptinin, and 1 µM E64), phosphatase inhibitors (2 mM Na_3_VO_4_, 1 mM NaF, 2 mM Na_4_O_7_P_2_, and 2 mM βGP), and 10 mM N-ethylmaleimide (NEM). MC3T3-E1 cell monolayers were rinsed with cold PBS and harvested in the non-denaturing buffer containing protease and phosphatase inhibitors and NEM. The samples were sonicated (Branson 450 Digital Sonifier, Danbury, CT, USA) and centrifuged at 13.4× *g* for 10 min at 4 °C. Supernatants containing whole cell lysates were analyzed for protein concentrations via BCA assay (Thermo Fisher Scientific, Waltham, MA, USA). Proteins were separated in polyacrylamide gels containing 10% SDS and transferred to nitrocellulose (Bio-Rad, Hercules, CA, USA). Following the transfer, the membrane was blocked in 4% nonfat milk in 5 mM Tris-Cl, pH 8.0 with 150 mM NaCl, and 0.1% Tween-20 (TBS-T) for 1 h at room temperature. The membranes were incubated with primary antibodies (Table 2), and the results were visualized with HRP-conjugated secondary antibodies (Jackson ImmunoResearch Laboratories, West Grove, PA, USA) and enhanced chemiluminescence (Thermo Fisher/Pierce, Rockford, IL, USA). Equal loading was determined with MemCode staining.

### 4.6. Staining and Image Quantification

#### 4.6.1. H&E Staining, Imaging, and Quantification of iWAT

Subcutaneous adipose tissue (iWAT) was fixed in 10% neutral buffered formalin for 24 h at 25 °C, embedded in paraffin, sectioned onto 5 µm sections, and hematoxylin and eosin (H&E) stained. The stained tissue was imaged using NanoZoomer software version 2.3.1 (Hamamatsu, Japan). Whole slide images were imported and analyzed using Visopharm VIS software version 2032.01 (Hørsholm, Denmark). Adipocyte size and number were quantified by thresholding for cell membranes; structures between 100 and 40,000 µm^2^ with a circularity index of 3.5 or less were considered adipocytes.

#### 4.6.2. Cell Culture Staining

Alkaline-phosphatase-positive cells were visualized using an alkaline phosphate staining kit (Sigma-Aldrich, Burlington, VT, USA), collagen development was detected using Picrosirius Red solution (Abcam, Cambridge, UK), and mineralization was detected using 5% silver nitrate (Sigma-Aldrich) in water (Von Kossa staining). Alkaline phosphatase and Picrosirius Red stains were conducted on day 6 for all experiments, and Von Kossa stains were conducted on days 10–16 depending on calcification. For all stains, media were removed and replaced with 10% neutral buffered formalin. After 12 min of incubation, the cells were rinsed with de-ionized water. Staining solutions (500 µL/well) were added and incubated for 30 min at room temperature followed by another de-ionized water rinse. Alkaline phosphatase and collagen stain incubation were carried out without light exposure, and Von Kossa stains were carried out under UV light exposure. 

#### 4.6.3. Image Capture and Quantification

Plates were allowed to dry completely prior to imaging. All images were captured using an iPhone 12 mini 1× back-facing camera (Apple, Cupertino, CA, USA) and the Camera+ app version 23.24 (LateNightSoft, Madrid, Spain). Plates were placed on a lightbox (Artograph), and the camera was placed at a set distance from the lightbox. Individual images were obtained as TIFF files for each well at a set focus, exposure, and color temperature for each cell culture plate. For quantification, images were opened in GNU Image Manipulation Program (GIMP, Charlotte, CA, USA), and clipped. The clip area was determined for each plate by choosing a size that would only include the well area but also exclude as much cell layer peeling as possible. Clipped images were then re-combined into a single image to include all experimental wells within a plate, flattened, and exported as a TIFF file. Images were re-opened in FIJI (FIJI is just ImageJ) and converted to an 8-bit, grayscale, inverted image. A region of interest was placed around each individual well. The thresholding function was used to exclude backgrounds not associated with the well, and a mean gray value for each region of interest was acquired using the measure function.

### 4.7. Statistical Analysis

Data were analyzed in GraphPad Prism version 9. Specifics for statistical testing are described in figure captions. We used an unpaired *t*-test, ANOVA, or two-way ANOVA depending on the data being analyzed. Post hoc tests for ANOVA or two-way ANOVA were either Dunnet’s test for multiple comparisons to control (within the day after two-way ANOVA) or Šídák’s multiple comparison test when there were only two groups per family (day). Alpha was set to 0.05 for statistical significance.

## Figures and Tables

**Figure 1 ijms-24-13423-f001:**
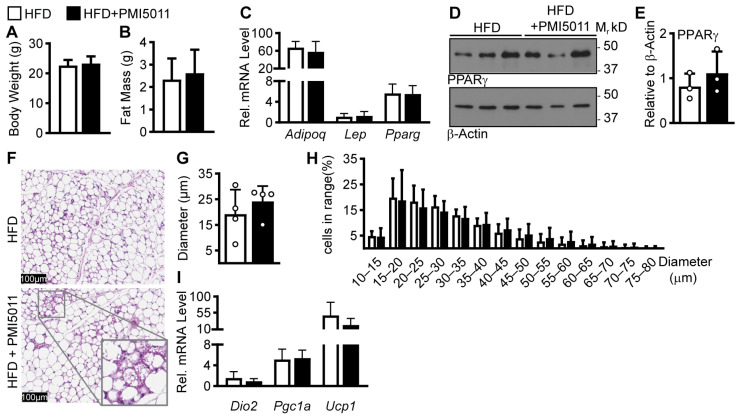
Effects of a 12-week HFD diet on adipose tissue with or without PMI5011 supplementation. (**A**,**B**). Average body weight and fat mass in female mice on an HFD with (n = 12) or without (n = 14) PMI5011. (**C**). Relative mRNA levels for adipogenic genes in iWAT acquired from a subset of the cohort (n = 5, HFD and 6, HFD + PMI5011) shown in (**A**,**B**). (**D**,**E**). Western blot of PPARγ from iWAT and the relative abundance was quantified and normalized to β-Actin (n = 3 for both groups). (**F**). (**E**,**H**) staining of iWAT tissue slices for adipocyte visualization; magnified section shows multilocular cells. (**G**,**H**). Average adipocyte diameter and adipocyte size histogram for mice on an HFD with or without PMI5011 (n = 4 for each group). (**I**). Relative mRNA levels for thermogenic genes in iWAT acquired from a subset of the cohort (n = 5, HFD and 6, HFD + PMI5011) shown in (**A**,**B**). No significant differences were found for any parameter (*t*-test, α = 0.05).

**Figure 2 ijms-24-13423-f002:**
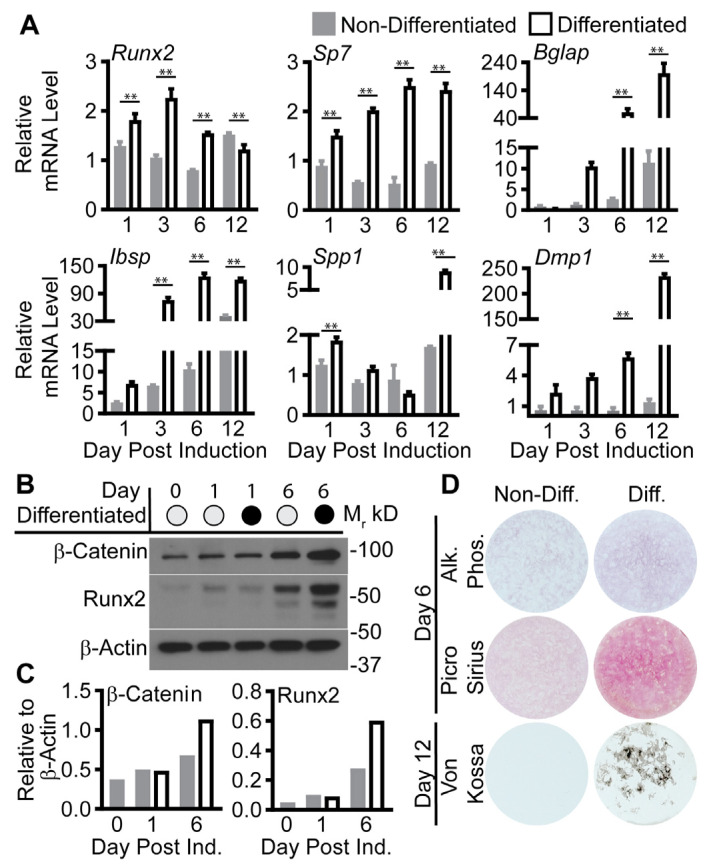
MC3T3-E1 pre-osteoblast model efficacy. (**A**). Relative mRNA levels of major osteogenic transcription factors (*Sp7* and Runx2) and genes that encode for extracellular proteins (*Bglap, Ibsp, Spp1,* and *Dmp1*) from non-differentiated and differentiated cells collected at 1, 3, 6, and 12 days (n = 3 technical replicates per group from a representative experiment). (**B**,**C**). Western blots for β-catenin, Runx2, and the loading control, β-actin, at days 0, 1, and 6 of differentiation. (**D**). Representative alkaline phosphatase and Picro Sirius staining for collagen on day 6 and Von Kossa staining for mineralization on day 12. Statistical analysis of relative mRNA levels used two-way ANOVA followed by Šídák’s multiple comparison test for differences within the day. Day and Treatment effects were *p* < 0.001 for all genes. (α = 0.05, ** indicates *p* < 0.01; open (−) or closed (+) circle = − or + indicated condition).

**Figure 3 ijms-24-13423-f003:**
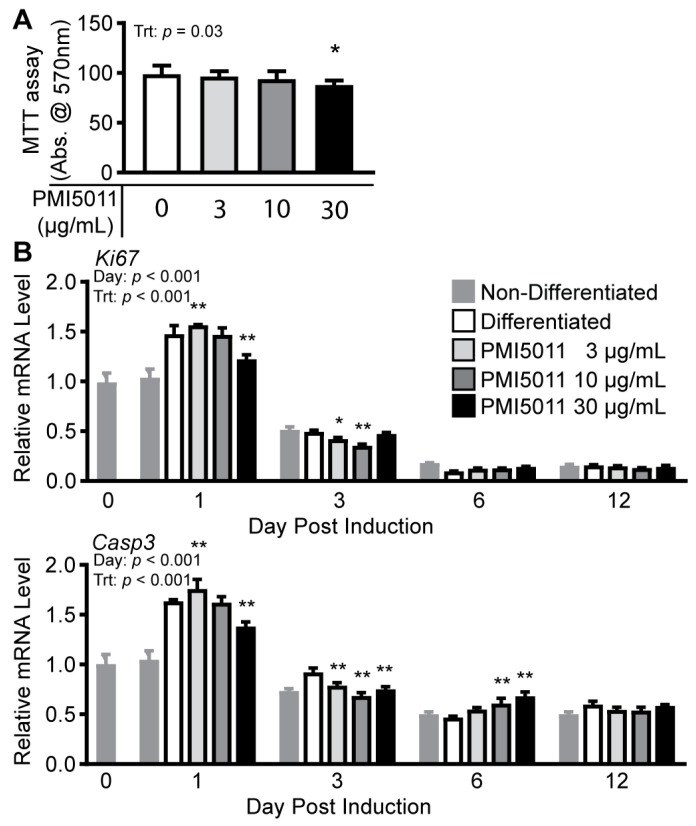
Effect of PMI5011 on proliferation and apoptosis. (**A**). MMT assay for MC3T3-E1 cells plated in MEMa without ascorbic acid for 24 h prior to supplementation with PMI5011 at increasing concentrations for 48 h. (**B**). Relative mRNA for proliferation gene *Ki67* and apoptosis gene *Casp3* during differentiation. Statistical analysis was conducted using one-way ANOVA (MMT assay) or two-way ANOVA (mRNA levels), and multiple comparisons were made to control (PMI5011 0 µg/mL) for MTT assay or differentiated within the day for mRNA levels using Dunnett’s post hoc test (α = 0.05, * and ** indicate *p* < 0.05 or *p* < 0.01, respectively).

**Figure 4 ijms-24-13423-f004:**
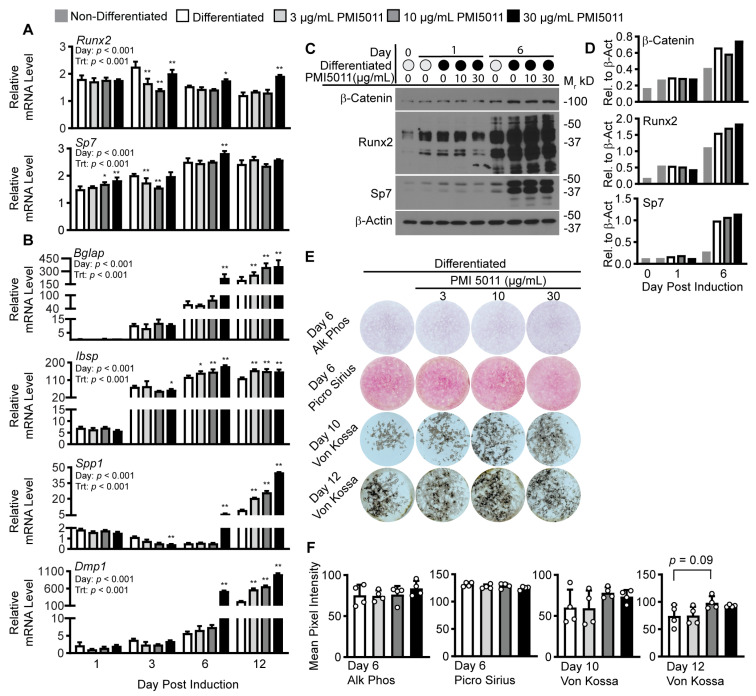
Effect of PMI5011 on in vitro osteogenesis. (**A**,**B**). Relative mRNA levels for major osteogenic transcription factors (*Sp7* and *Runx2*) and non-collagenous osteogenic proteins (*Bglap, Ibsp, Spp1,* and *Dmp1*) from differentiated cell lysates treated with increasing concentrations of PMI5011 collected at indicated days post-induction for differentiated cells (n = 3 technical replicates per group from a representative experiment of 3 independent experiments). (**C**,**D**). Western blots for β-catenin, RUNX2, SP7, and β-actin (β-Act, loading control), on days 0, 1, and 6 post inductions. (**E**). Staining outcomes for alkaline phosphate and collagen on day 6 and mineralization on days 10 and 12 in cells differentiated with or without increasing concentrations of PMI5011 (n = 4 wells per group). (**F**). Bar graphs comparing group differences in staining outcomes (n = 4 wells per treatment). Statistical analysis of relative mRNA levels used two-way ANOVA followed by Dunnet’s multiple comparison tests for differences within the day compared to control (differentiated). Statistical analysis of staining outcomes used one-way ANOVA followed by Dunnet’s multiple comparison test to control (differentiated). (α = 0.05, * and ** indicate *p* < 0.05 or *p* < 0.01, respectively, open (−) or closed (+) circle = − or + indicated condition).

**Figure 5 ijms-24-13423-f005:**
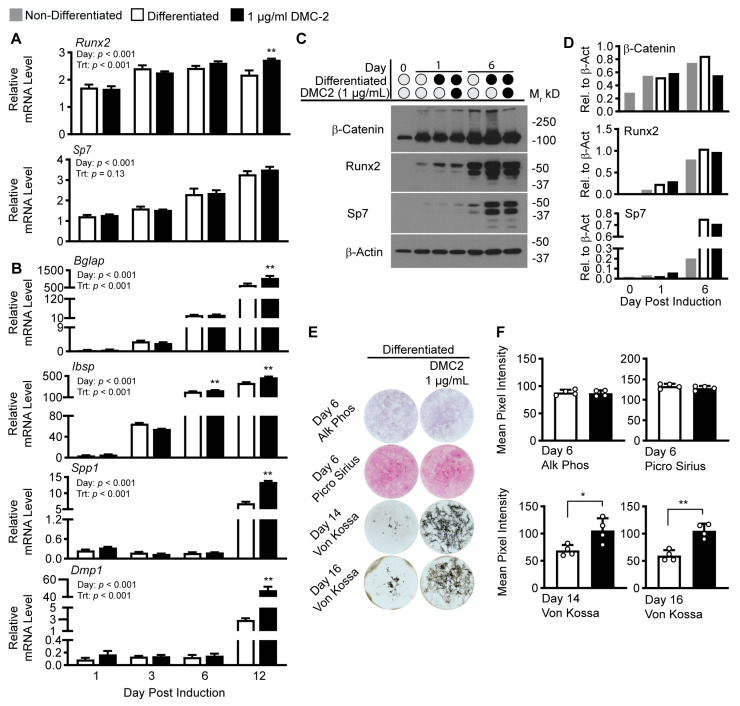
Effect of DMC-2 on osteoblast differentiation and activity. (**A**,**B**). Relative mRNA levels for major transcription factors (*Sp7* and *Runx2*) and non-collagenous proteins (*Bglap, Ibsp, Spp1,* and *Dmp1*) from differentiated cell lysates treated with increasing concentrations of DMC-2 collected on indicated days post induction for differentiated cells (n = 3 technical replicates per group from a representative experiment of three independent experiments). (**C**,**D**). Western blots for β-catenin, RUNX2, SP7, and β-actin (β-Act, loading control), on days 0, 1, and 6 of differentiation. (**E**). Staining outcomes for alkaline phosphate and collagen on day 6 and mineralization on days 14 and 16 in non-differentiated cells or cells differentiated with or without increasing concentrations of PMI5011 (n = 4 wells per group). (**F**). Bar graphs comparing group differences in staining outcomes (n = 4 wells per treatment). Statistical analysis of relative mRNA levels used two-way ANOVA followed by Šídák’s multiple comparison test for differences within the day. Statistical analysis of staining outcomes used unpaired t-tests to compare cells differentiated with and without DMC-2. (α = 0.05, * and ** indicate *p* < 0.05 or *p* < 0.01, respectively, open (−) or closed (+) circle = − or + indicated condition).

**Table 1 ijms-24-13423-t001:** Gene list and primer sequences.

Gene ID	Gene Name	Accession #	Sequence–Forward	Sequence–Reverse
*Ppib*	Peptidulprolyl isomerase B (cyclophillin B)	NM_011149.2	5′-TCC ATC GTG TCA TCA AGG ACT T-3′	5′-CTG ATC TGG GAA GCG CTC A-3′
*Adipoq*	Adiponectin	NM_009605.5	5′-CAT GCC GAA GAT GAC GTT ACT A-3′	5′-ACG CTG AGC GAT ACA CAT AAG-3′
*Lep*	Leptin	NM_008493.3	5′-GGC TTT GGT CCT ATC TGT CTT ATG-3′	5′-CCG ACT GCG TGT GTG AAA T-3′
*Pparg*	Peroxisome proliferator activated receptor gamma	NM_011146	5′-ATG TCA GTA CTG TCG GTT TCA G-3′	5′-AGC GGG AAG GAC TTT ATG TAT G-3′
*Dio2*	Iodothyronine deiodinase 2	NM_010050.4	5′-CTC CTA GAT GCC TAC AAA CAG G-3′	5′-GCA CTG GCA AAG TCA AGA AG-3′
*Pgc1a*	Peroxisome proliferator-activated receptor gamma coactivator 1-alpha	NM_008904	5′-AGC CTC TTT GCC CAG ATC TTC-3′	5′-CCA TCT GTC AGT GCA TCA AAT GA-3′
*Ucp1*	Uncoupling protein 1	NM_009463.3	5′-GAT CTT CTC AGC CGG AGT TT-3′	5′-GCC TTC ACC TTG GAT CTG AA-3′
*Rlp13a*	Ribosomal protein L13a	NM_009438.5	5′-CCA AGA TGC ACT ATC GGA AGA A-3′	5′-CTT GAG GAC CTC TGT GAA CTT G-3′
*Runx2*	Runt related transcription factor 2	NM_001146038	5′-CCA GGC GTA TTT CAG ATG ATG A-3′	5′-GTC CTC AGT GAG GGA TGA AAT G-3′
*Sp7*	Transcription factor 7	NM_130458.4	5′-CTC CTC GGT TCT CTC CAT CT-3′	5′-GGA GCC ATA GTG AGC TTC TTC-3′
*Bglap*	Bone gamma-carboxyglutamate protein (osteocalcin)	NM_007541.3	5′-CCA AGC AGG AGG GCA ATA A-3′	5′-TCG TCA CAA GCA GGG TTA AG-3′
*Ibsp*	Intergrin binding sialoprotein (bone sialoprotein)	NM_008318.3	5′-CTC CAC ACT TTC CAC ACT CTC-3′	5′-TTT CTG CAT CTC CAG CCT TC-3′
*Spp1*	Secreted phosphoprotein 1 (ostepontin)	NM_001204201.1	5′-GGC TGA ATT CTG AGG GAC TAA C-3′	5′-GCC ATG TGG CTA TAG GAT CTG-3′
*Dmp1*	Dentin matrix protein	NM_001359013.1	5′-CAG AGG GAC AGG CAA ATA GTG-3′	5′-GTC GTC TTC ATC ATC CTC CTT ATC-3′
*Ki67*	Antigen identified by monoclonal antibody Ki67	NM_001081117.2	5′-GAT TCC ATT AAC AAG AGT GAG GGA-3′	5′-CTG TGA GTG CCA AGA GAC TTC-3′
*Casp3*	Caspase 3	NM_001284409.1	5′-GCA GCT TTG TGT GTG TGA TTC-3′	5′-GCA GGC CTG AAT GAT GAA GA-3′
All Primers designed and purchased from Integrated DNA Technologies (Coralville, IA, USA)				

**Table 2 ijms-24-13423-t002:** Antibody information.

Anti-:	Host	Manufacturer	Catalogue #
PPARγ	Mouse Monoclonal	Santa Cruz (Santa Cruz, CA, USA)	SC-7273
β-Actin	Rabbit Polyclonal	GeneTex (Irvine, CA, USA)	GTX109639
β-Catenin	Rabbit Polyclonal	Cell Signaling (Danvers, MA, USA)	9562
Runx2	Mouse Monoclonal	Santa Cruz (Santa Cruz, CA, USA)	SC-390351
Sp7	Rabbit Monoclonal	Abcam (Cambridge, UK)	AB209484

## Data Availability

The data presented in this study are available within the manuscript.

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
