# Peer review of "Extract of Artemisia dracunculus L. Modulates Osteoblast Proliferation and Mineralization"

_ijms, 2023, doi:10.3390/ijms241713423_

Round 1

Reviewer 1 Report

I read with interest this manuscript focusing on the effects of Artemisia dracunculus L. on osteogenesis, whose eventual positive effects could push research on verify their use a treatment for improving insuline sensitivity and blood glucose via skeletal muscle without negative effects on bone.  

I found the study solid, well designed, and overall interesting. The manuscript is very well written and explains rigorously the methodology used and the results obtained. I congratulate with the authors for their research. 

I have only a comment:  

please explain better the sentence reported in rows 121-122 in order to make clearer results related to changes in MTT assay absorbance.

Author Response

Thank you for the careful review and the comment. We agree that the sentence needs clarifying.  The sentence (L119-121) now reads “The 12% decrease in absorbance we observed with 30 µg/ml PMI5011 indicates fewer viable cells but cannot differentiate between cytotoxicity, proliferation or apoptosis [30,31].  We deleted the phrase “and may reflect early modulation of ostengenesis”.

Reviewer 2 Report

This article reports that  an extract of Artemisia dracunculus L., called PMI5011, improves blood glucose and insulin sensitivity. One of the  active compounds in the extract was identifyed as 2’,4’-dihydroxy-4-methoxydihydrochalcone (DMC-2). The effect is mediated via skeletal muscle. The hypoglycemic effect is compared to thiazolidinediones (TZDs), that improve insulin sensitivity via action on adipocytes. However, TZDs degrades bone by inhibiting osteoblasts. Interestingly, both the plant extract and  1µg/m DMC-2 significantly increase osteogenic proteins and cell culture mineralization. Altogether, data suggest that Artemisia dracunculus have the potential to control glycemia and simultaneously promoting bone health.  

Some of the effects of Artemisia dracunculus L were already known, but they are correctly referenced at the introduction and discussion. The study is interesting, and the methods are appropriate. Biomarkers used are correct and numerous. Results are clear and well described in panels and auto explicatory figure legends. Statistical analyses are also correct. In summary, I find the article suitable for publication once the following point will be addressed.

The relative content of DMC-2 in the raw ethanolic extract of tarragon (A. dracunculus) would be determined and discussed in relation to other possible active compounds. Ref. 21 states that the content is less that 3%, but there is not analysis in this work and the purity of the chemical is unknown. On the other hand, it is stated at lines 333-334 that DMC-2 was synthesized as previously described [60], but according to my consult, ref. 60 purified DMC-2, but I could not find any synthesizing protocol. Please, clarify this point

Author Response

Thank you for the careful review and the comment. In the current study, we are using PMI5011 from the same PMI5011 lot extracted from Russian tarragon as used in the studies described in reference 21. To clarify this point, Section 4.1 of Materials and Methods (L330-337) is now edited to read “The PMI5011 ethanolic extract from A. dracunculus was provided by the Botanical and Dietary Supplement Research Center at Pennington Biomedical Research Center. The PMI5011 ethanolic extract used in the current study is from the same lot as previously described (21). The bioactive DMC-2 is 1.76% (w/w) of the total extract.

We further edited Section 4.1 to clarify that “Synthetic DMC-2 (2′, 4′–dihydroxy-4-methoxydihydrochalcone) was produced by custom synthesis to 99% purity by Gateway Biochemical Technology, Inc as previously noted [40]. Note: We referenced the paper previously numbered 60 earlier in the paper (L221) to be clear that we are using synthesized DMC-2, so the reference numbers have been adjusted.